# Increased Production of Inflammatory Cytokines after Inoculation with Recombinant Zoster Vaccine in Mice

**DOI:** 10.3390/vaccines10081339

**Published:** 2022-08-18

**Authors:** Tetsuo Nakayama, Akihito Sawada, Takeshi Ito

**Affiliations:** 1Laboratory of Viral Infection, Ömura Satoshi Memorial Institute, Tokyo 108-8641, Japan; 2Department of Pediatrics, Kitasato University Hospital, Sagamihara 252-0329, Kanagawa, Japan

**Keywords:** inflammatory cytokine, G-CSF, IL-6, Th2 cytokine, recombinant zoster vaccine

## Abstract

Increasing numbers of patients with zoster were reported recently, and recombinant zoster vaccine (Shingrix^®^) was licensed using the AS01_B_ adjuvant system. Although it induces highly effective protection, a high incidence of local adverse events (regional pain, erythema, and swelling) has been reported with systemic reactions of fever, fatigue, and headache. To investigate the mechanism of local adverse events, cytokine profiles were investigated in mice injected with 0.1 mL of Shingrix^®^. Muscle tissue and serum samples were obtained on days 0, 1, 3, 5, and 7, and at 2 and 4 weeks after the first dose. The second dose was given 4 weeks after the first dose and samples were obtained on days 1, 3, 5, 7, and 14. IL-6 and G-CSF were detected in muscle tissues on day 1 of the first injection, decreased on day 3 and afterward, and enhanced production was demonstrated on day 1 of the second dose. In sera, the elevated levels of IL-6 were detected on day 1 of the first dose, and IL-10 was detected on day 1 with increased levels on day 3 of the first dose. IL-4 was detected in muscle tissue on day 1 of the second dose and IL-5 on day 1 of both the first and second doses. IFN-γ production was not enhanced in muscle tissue but increased in serum samples on day 1 of the first dose. These results in the mouse model indicate that the induction of inflammatory cytokines is related to the cause of adverse events in humans.

## 1. Introduction

Varicella zoster virus (VZV) is a causative agent of chickenpox in childhood, resulting in latent infection in the dorsal sensory ganglia of the spinal cord. In the elderly, when cellular immunity decreases, reactivated VZV migrates to the dermatome related to the affected spinal cord region and causes herpes zoster. The typical cutaneous region of acute herpes zoster includes clusters of vesicles causing serious pain and itching [1,2]. Some patients develop post-herpetic neuralgia, and their daily lives are disturbed [3,4]. Recent developments in treatment with chemotherapy, radiation, and surgical approaches, including stem cell transplant for malignant diseases have contributed to prolonged life expectancy [5]. Live attenuated varicella zoster vaccine was licensed in 1995 in the US, and the epidemiology of chickenpox was renewed to decrease the number of patients in young generations [6]. Although individuals susceptible to chickenpox are decreasing in the population, the number of patients with zoster is increasing due to the reduced chances of booster reaction among young adults, as well as the elderly [7,8].

Live attenuated varicella vaccine reduced the disease burden of herpes zoster by 61.1%, post-herpetic neuralgia by 66.5%, and incidence of herpes zoster by 51.3% [9], but the administration of live zoster vaccine is prohibited in immune-compromised subjects. The recombinant zoster vaccine (Shingrix^®^) containing purified gE protein was licensed as inactivated vaccine. The AS01_B_ adjuvant system is used to enhance immunogenicity. It showed high vaccine effectiveness over 95% but the incidence of local adverse events was 6–7 times higher than that of a placebo control [10,11]. The most common symptom was pain at the injection site and systemic adverse reactions were fatigue, myalgia, and headache. Less than 10% of the subjects experienced grade 3 reactions, but they resolved within 3 days [10]. Now, Shingrix^®^ is recommended for subjects ≥50 years of age, but they are worried about serious regional pain at the injection site [10,11,12]. It was considered as a reaction caused by the AS01_B_ adjuvant system, using monophosphoryl lipid A (MPL), *Quillaja Saponaria Molina* extract QS-21, cholesterol, and dioleoylphosphatidylcholine (DOPC) in the composition of liposomes [13]. MPL is used as an AS04 adjuvant in HPV Cervarix^®^, which was related to the adverse event of acute local pain. Some HPV recipients complained of chronic systemic pain, headache, and unexplained functional autonomic nervous disorders [14]. These were not always related to the vaccination because a similar incidence of chronic pain or these symptoms was noted among unvaccinated subjects [15]. We previously reported that inflammatory nodules were observed after immunization with adjuvanted vaccines in mice; inflammatory cytokines were detected from 3 to 48 h after immunization and enhanced production was noted after the second dose [16].

The purpose of the present study is to investigate the cytokine profiles in a mouse model to elucidate the adverse events associated with local pain after immunization with Shingrix^®^.

## 2. Materials and Methods

### 2.1. Study Plan

Female BALB/c mice aged four weeks were purchased from Charles River Laboratory Japan. Ten mice were injected with 0.1 mL physiological saline for control and muscle tissue and serum samples were obtained from five mice for each on days 1 and 3. In preliminary experiments, 36 mice were injected with 0.1 mL of Shingrix^®^ adjuvanted vaccine in the left thigh muscle. Muscle tissues and serum samples were collected from three mice at each time-point on days 0, 1, 3, 5, and 7, and at 2 and 4 weeks after the first dose. The second dose was given at 4 weeks after the first dose, and the following samples were obtained in a similar fashion. An increased concentration of cytokines was detected on days 1 and 3 of the first and second doses. Then, we performed the additional experiment, using three mice for each before inoculation and on days 1 and 3 after the first and second doses. Six mice were used before and on days 1 and 3 following first and second injections, and three at the other time-points were analyzed. Thigh muscle tissue was obtained after anesthesia with pentobarbital sodium throughout the experiment time-points and blood samples were obtained through cardiac puncture at the same time. Samples were applied for the detection of cytokines. The ethical committee of animal research of Kitasato University (approval No. 20-016) approved the study protocol.

### 2.2. Cytokine Measurement

Muscle tissues were dissected to obtain approximately 150 mg and cut into small pieces and homogenized in 1.0 mL RPMI, supplemented with 1% protease inhibitor (Nacalai Tesque, Kyoto, Japan) using Precellys Lysing Kits (BERTIN Corp., Rockville, MD, USA). The muscle homogenate was centrifuged, filtrated through a 0.45 μm filter, and stored at −80 °C until assay. They were assayed in the same plate. GM-CSF, IFN-γ, IL-2, IL-4, IL-5, IL-10, IL-12, and TNF-α were measured using the Bio-Plex mouse cytokine panel (Bio-Plex, BIO-RAD Laboratories, Hercules, CA, USA). G-CSF and IL-6 were, respectively, assayed using mouse G-CSF and mouse IL-6 SimpleStep ELISA^®^ Kits (Abcam, Cambridge, UK).

Stocked serum samples were also subjected to cytokine assays.

### 2.3. Statistical Analysis

Cytokine concentrations were analyzed in box-and-whisker plots with the median titer and lower and upper range of 5–95%. Significance (*p* < 0.05) was determined by the Mann–Whitney U test, using BellCurve for Excel (Social Survey Research Information Co., Ltd., Tokyo, Japan).

## 3. Results

### 3.1. Inflammatory Cytokine Production in Muscle Tissues and Serum Samples

The cytokine concentration is shown as the mean ± 1.0 SD. IL-6 and G-CSF profiles in muscle tissues are shown in Figure 1. IL-6 was detected at low levels in muscle tissue (28 ± 17 pg/mL) before injection, increased on day 1 after the first dose (176 ± 83 pg/mL, *p* < 0.01) and decreased on day 3 (59 ± 33 pg/mL). Higher levels of IL-6 (575 ± 292 pg/mL, *p* < 0.01) were detected on day 1 of the second dose. A low level of IL-6 (19 ± 22 pg/mL) was detected as a control on day 1 of physiological saline injection.

G-CSF (773 ± 321 pg/mL) was detected in muscle tissue on day 1 of the first dose, decreased to undetectable levels (<20 pg/mL) on day 3 and afterwards, and a high concentration of G-CSF (806 ± 235 pg/mL) was observed on day 1 after the second dose. A low level of G-CSF (33 ± 23 pg/mL) was detected as a control on day 1 of physiological saline injection.

Serum IL-6 was also investigated in serum samples, shown in Figure 2. IL-6 was detected before injection (82 ± 46 pg/mL). A higher level of IL-6 was detected on Day 1 (153 ± 82 pg/mL) but was not significant after the first dose and did not increase after the second dose. A low level of IL-6 (65 ± 22 pg/mL) was detected in serum samples as a control on day 1 of physiological saline injection. The elevated levels of G-CSF (>1200 pg/mL) were detected in serum samples as a control on day 1 of physiological saline injection (data not shown).

IL-10 was detected in serum samples on day 1 (11.8 ± 10.6 pg/mL), increased on day 3 (36.6 ± 13.1 pg/mL, *p* = 0.07), and decreased on day 5 and later after the first dose. Lower levels of IL-10 (12.8 ± 9.11 pg/mL) were detected on day 1 after the second dose. IL-10 was detected in the serum sample (6.18 ± 3.31 pg/mL) after physiological saline injection.

### 3.2. IL-5 and IFN-γ Profiles in Muscle Tissue and Serum Samples

IL-5 and IFN-γ profiles are shown in Figure 3. IL-5 was detected at a low level (14.77 ± 18.46 pg/mL) on day 1 after the first dose and at a similar level (15.49 ± 5.16 pg/mL) on day 1 after the second dose in muscle tissue. Serum IL-5 increased on day 1 after the first dose (14.77 ± 18.46 pg/mL) and on day 1 after the second dose (15.50 ± 5.11 pg/mL). Less than 2 pg/mL of IL-5 was detected in muscle tissue and IL-5 (6.30 ± 2.77 pg/mL) was detected in serum samples on day 1 of physiological saline injection.

The baseline IFN-γ was 80.52 ± 31.47 pg/mL in muscle tissues, decreased on day 1 (40.47 ± 6.89 pg/mL) after the first dose, and recovered two weeks later. After the second dose, a similar profile was observed. Increased IFN-γ production in serum samples was observed on day 1 after the first dose (24.55 ± 8.28 pg/mL, *p* < 0.05), in comparison with before the injection (6.81 ± 8.47 pg/mL) and decreased on day 3 and afterwards after the first dose. Less than 2 pg/mL of IFN-γ was detected in muscle tissue and serum samples on day 1 of physiological saline injection.

### 3.3. IL-2, IL-4, TNF-α, and IL-10 in Muscle Tissues

The Th1 cytokine profile of IFN-γ in muscle tissues is shown in Figure 3. Following the injection of Shingrix^®^, other cytokine profiles are shown in Figure 4. The baseline IL-2 level was 17.15 ± 4.53 pg/mL, which decreased after immunization until day 7 after the first dose. It recovered within 2–4 weeks later. After the second dose, IL-2 decreased during 2 weeks. Similar profiles were observed for the TNF-α and IL-10. GM-CSF and IL-12 showed similar profiles (data not shown).

However, a different IL-4 Th2 cytokine profile was observed; an increased level of IL-4 was detected in muscle tissues obtained from mice injected with the second dose of Shingrix^®^ (9.83 ± 7.79 pg/mL).

## 4. Discussion

A vaccine induces acquired specific immune responses of antibody and cell-mediated immunity through immunological reactions. Innate immunological reactions are mediated by pathogen-associated molecular patterns (PAMPs) and damage-associated molecular patterns (DAMPs) [17,18]. Shingrix^®^ contains recombinant gE protein of varicella zoster virus, encapsulated into the ASO1_B_ liposome adjuvant system. AS01_B_ contains MPL, QS-21, cholesterol, and DOPC. MPL is a component of AS04 combined adjuvant with aluminum salt and acts as a ligand to TLR4 of PAMPs, inducing inflammatory cytokines, IL-1β, IL-6, and tumor necrosis factor (TNF)-α [19]. They act to promote inflammatory reactions after vaccination, and acquired immune responses developed through such inflammatory reactions. From another point of view, this process reflects the adverse reactions of local pain and fever [17]. The human cytokine profile was examined after immunization with trivalent influenza vaccine, and IP-10 and IFN-γ were detected at 7 h, with a peak response between 16 and 24 h after immunization, demonstrating distinct early cytokine profiles of IL-6 and other cytokines in subjects with myalgias [20]. IL-6 is a critical mediator of acute febrile inflammation, and binding to soluble IL-6 receptor promotes the recruitment of leukocytes to inflammatory sites through signaling with the multiple participation of cytokines and chemokines [21,22]. In our previous study of cytokine profiles in infants immunized with DPT, Hib, and PCV, significantly elevated levels of G-CSF were detected in serum, obtained from those with a febrile illness, and we considered G-CSF to be an indicator of adverse reactions [23]. No significant difference in the incidence of solicited local adverse events after doses 1 and 2 of Shingrix^®^ was observed in a clinical study [24] but there has been no report on the relationship between adverse reactions and inflammatory cytokines.

IL-6 and G-CSF were detected in muscle tissue on day 1 after the first dose, decreased on day 3, and higher levels of inflammatory cytokines were detected on day 1 of the second dose. There is some possibility that the very short-term elevation of inflammatory cytokine production was influenced by the traumatic effects of the large-volume injection because of the 50 µL volume limit for intramuscular injections in mice [25]. We examined cytokine production in mice injected with 100 μL physiological saline. Low levels of IL-6 (19 ± 22 pg/mL) and G-CSF (33 ± 23 pg/mL) were detected as a control of day 1 of physiological saline injection. Only serum G-CSF was elevated on day 1 of physiological saline injection.

In skin lesions, antigen-presenting cells (dendritic cells) are abundantly located as sentinel immunological cells to regulate immune responses with cytokine productions. In our previous mouse experiments, an inflammatory nodule was observed after injection with adjuvanted vaccines, containing neutrophils and macrophages [16]. Neutrophils migrated in the injection site and induced neutrophil extracellular traps (NETs) by reactive oxygen series and cellular DNA-stimulated DAMPs [16]. Markedly elevated levels of G-CSF were detected in muscle tissues in the early phase on day 1 of immunization after the first and second administrations of Shingrix^®^, similar to the finding after immunization with adjuvanted HPV vaccines [16]. G-CSF is produced by vascular endothelial cells, macrophages, and immunocompetent cells and is considered to represent neutrophil maturation. Recently, the immune function of G-CSF was postulated to be based on T-cell function through the modulation of endogenous cytokines. Rutella et al. [26] reported that CD4+ cells activated by G-CSF produced elevated levels of IL-10 but undetectable levels of IL-2 and IL-4 in healthy subjects after G-CSF administration. IL-10 is a cytokine that suppresses the inflammatory reactions [27].

In the present study, IL-4 and IL-5 were increasingly observed on day 1 after the second dose in muscle tissue (Figure 3 and Figure 4). IL-4 is an important Th2 cytokine that modulates the differentiation of Th2 cellular populations. IL-5 has activity to induce the maturation of basophilic cells and B cells to plasma cells to produce immunoglobulin. Didierlaurent et al. [18] reported that the effects of AS01 are rapid and transient activation of CD4+ T-cell-mediated immune responses in injected muscle and draining lymph nodes and AS01 is efficient at promoting CD4+ T-cell-mediated immune responses. Although AS01 enhanced antibody production with an increased number of CD4+ T cells in peripheral blood, antigen-specific CD8+ T cells were low in peripheral blood, based on clinical experience with AS01-adjuvanted vaccine.

Different cytokine profiles are shown in Figure 4. IL-2, TNF-α, and IL-10 production in muscle tissues decreased after injection with Shingrix^®^ and recovered within two weeks, like the IFN-γ production profile in muscle tissue (Figure 3). Th1 and Th2 responses were reported in spleen cell cultures stimulated with gE protein using C57BL/6J mice [28]. They used C57BL/6J mice, known as Th1 dominant [29]. In the present study, we used BALB/c mice, being Th2-dominant, and we aimed to analyze the local innate immune responses. These might influence the decreased Th1 cytokines after injection with Shingrix^®^. Schülke [30] reviewed the pleiotropic regulatory effects of IL-10 on multiple cell types, resulting in limiting or terminating excess T-cell responses. High levels of G-CSF in sera on day 1 of saline injection may influence the cytokine networks.

Cytokine profiles of IL-6 and IL-10 time-courses in muscle tissues and sera reflect the regional and systemic adverse events. Grade 3 serious adverse events were observed in less than 10% of the subjects. When daily activities are disturbed, taking non-steroidal anti-inflammatory drugs will relieve the pain, suppressing the inflammatory cytokine IL-6 [31].

## 5. Conclusions

Shingrix^®^-containing AS01_B_ induced IL-6 and G-CSF in muscle tissues on day 1 of the injection in mice after the first and second doses. IL-6 was detected in serum samples on day 1 of the first dose, which can explain the local pain after immunization. IL-10 was induced in serum samples on days 1 and 3 of the first dose and would be related to the suppression of inflammatory responses.

## Figures and Tables

**Figure 1 vaccines-10-01339-f001:**
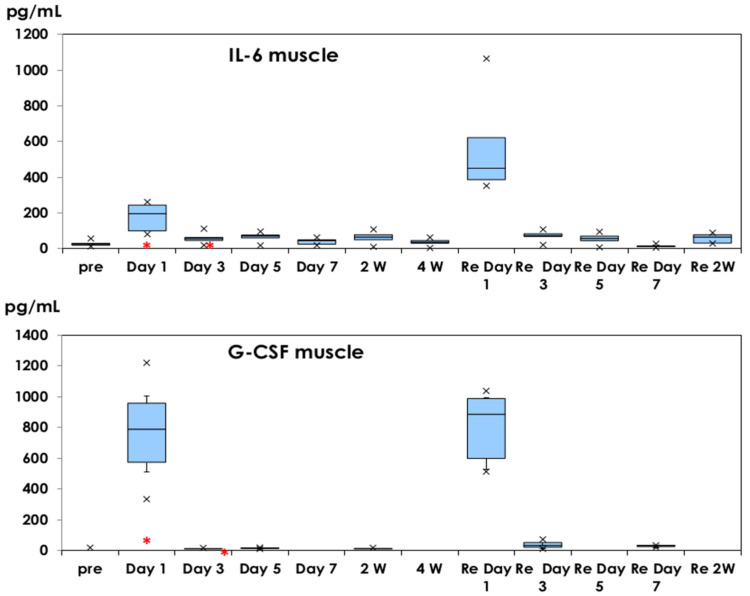
IL-6 and G-CSF production in muscle tissues. They are shown in box-and-whisker plots with median titers and the range of 5–95%. Samples were obtained before injection, on days 1, 3, 5, and 7, and at 2 and 4 weeks after the first dose. The second dose was administered at 4 weeks and samples were obtained based on the same schedule. *: mean concentration of five mice as a control on day 1 of physiological saline injection. ×: outliers.

**Figure 2 vaccines-10-01339-f002:**
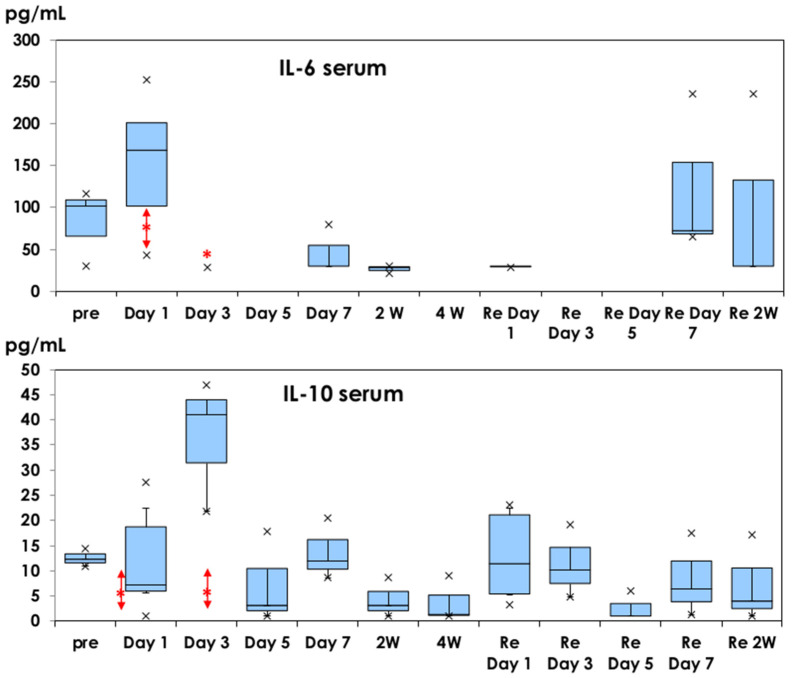
Serum IL-6 and IL-10 profiles. They are shown in box-and-whisker plots with median titers and the range of 5–95%. Samples were obtained before injection, on days 1, 3, 5, and 7, and at 2 and 4 weeks after the first dose. The second dose was administered at 4 weeks and samples were obtained based on the same schedule. *: mean concentration of five mice as a control on day 1 of physiological saline injection. 
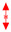
: mean ± 1.0 SD. ×: outliers.

**Figure 3 vaccines-10-01339-f003:**
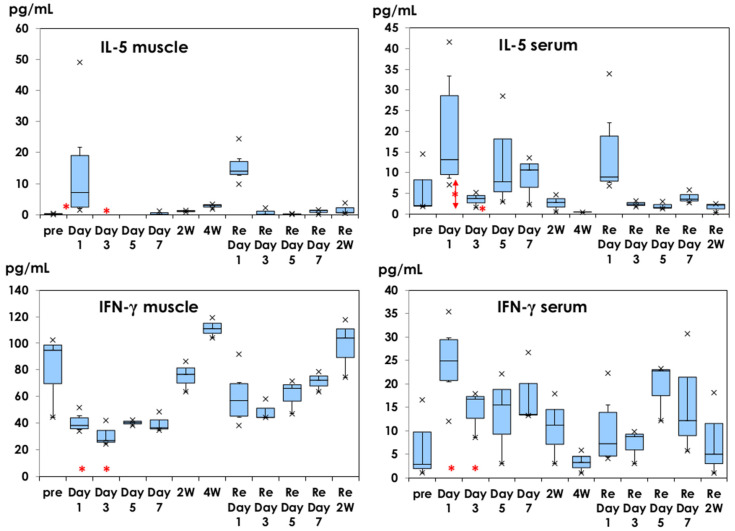
IL-5 and IFN-γ production in muscle tissues and serum samples. They are shown in box-and-whisker plots with a median titer and the range of 5–95%. Samples were obtained before injection, on days 1, 3, 5, and 7, and at 2 and 4 weeks after the first dose. The second dose was administered at 4 weeks and samples were obtained based on the same schedule. *: mean concentration of five mice as a control on day 1 of physiological saline injection. 
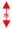
: mean ± 1.0 SD. ×: outliers.

**Figure 4 vaccines-10-01339-f004:**
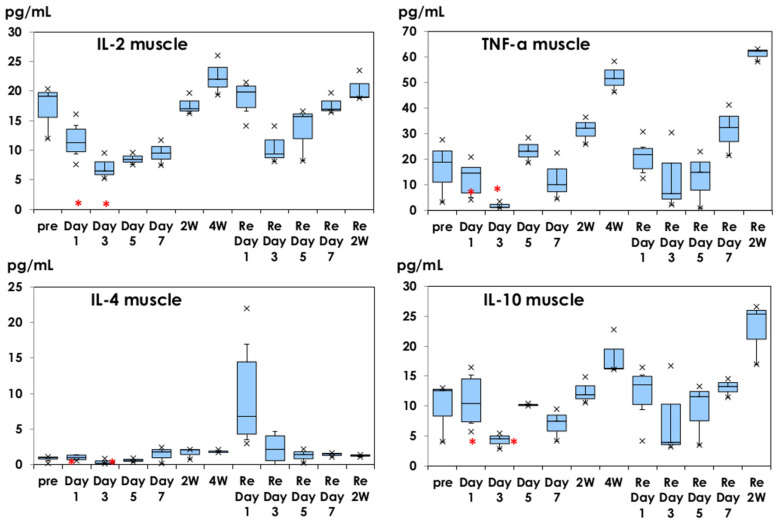
IL-2, IL-4, TNF-α, and IL-10 production in muscle tissues. They are shown in box-and-whisker plots with a median titer and the range of 5–95%. Samples were obtained before injection, on days 1, 3, 5, and 7, and at 2 and 4 weeks after the first dose. The second dose was administered at 4 weeks and samples were obtained based on the same schedule. *: mean concentration of five mice as a control on day 1 of physiological saline injection. ×: outliers.

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
