# Peer review of "Increased Production of Inflammatory Cytokines after Inoculation with Recombinant Zoster Vaccine in Mice"

_vaccines, 2022, doi:10.3390/vaccines10081339_

Round 1
Reviewer 1 Report
This revised manuscript improved some contents and explanations following reviewers' comments. However, one major issue is still there that the significance of this study is elusive. The motivation of this study is not fully presented, and it is still unclear what the overall finding of this manuscript compared to other studies. If the vaccination approach needs improvement, what will be the exact solutions? Another issue is the language which may cause misunderstanding during the review process. I would suggest the authors let a native English speaker proofread this manuscript.
Author Response
Thank you for the valuable comment.
【Comment】This revised manuscript improved some contents and explanations following reviewers' comments. However, one major issue is still there that the significance of this study is elusive. The motivation of this study is not fully presented, and it is still unclear what the overall finding of this manuscript compared to other studies. If the vaccination approach needs improvement, what will be the exact solutions? Another issue is the language which may cause misunderstanding during the review process. I would suggest the authors let a native English speaker proofread this manuscript.
【Response】I added the following sentences to describe the symptoms of adverse events (line 51-54): The most common symptom was pain at the injection site and systemic adverse reactions were fatigue, myalgia, and headache. Less than 10% of the subjects experienced grade 3 reactions, but they resolved within 3 days [10].
And last sentence of the Introduction was changed (line 66-68).
The purpose of the present study is to investigate the cytokine profile in a mouse model to elucidate the adverse events associated with local pain after immunization with Shingrix®.
As an actual solution for reducing the severity of symptoms, the following paragraph was added (line 255-258). Cytokine profiles of IL-6 and IL-10 time-courses in muscle tissues and sera reflect the regional and systemic adverse events. Grade 3 serious adverse events were observed in less than 10% of the subjects. When daily activities are disturbed, taking non-steroidal anti-inflammatory drugs will relieve the pain, suppressing the inflammatory cytokine IL-6 [30].
As the reviewer suggested, the revision was checked by a native researcher.
Reviewer 2 Report
Dear authors,
The manuscript improved a lot. You all showed the cytokine profile after injecting the Shingrix vaccine at the site of injection ( muscle) and in blood. But it is known that at the site of tissue damage, IL-6 is produced in high amount and it causes pain, and increase inflammation through positive feed back loop. In the discussion, you can suggest some anti IL-6 medications to administer together with the vaccines (suggested review- eg. https://doi.org/10.1016/j.immuni.2019.03.026; 10.2147/OARRR.S291388; https://doi.org/10.1016/j.bja.2017.11.096.) You can administer antagonist of IL-6 activation (soluble gp130) together with the vaccine and see whether it decreases the pain.
There are some corrections:
1. In the method section you have forgotten to mention that the serum samples were also analyzed for cytokines by ELISA
2. Line 224-is the word "until" appropriate?
3. In the introduction please mention how long the elders feel the pain after the vaccination of shingrix.
Thank you!
Author Response
Thank you for the valuable comments.
【Comment】The manuscript improved a lot. You all showed the cytokine profile after injecting the Shingrix vaccine at the site of injection (muscle) and in blood. But it is known that at the site of tissue damage, IL-6 is produced in high amount, and it causes pain, and increase inflammation through positive feed back loop. In the discussion, you can suggest some anti IL-6 medications to administer together with the vaccines (suggested review- e.g., https://doi.org/10.1016/j.immuni.2019.03.026; 10.2147/OARRR.S291388; https://doi.org/10.1016/j.bja.2017.11.096.) You can administer antagonist of IL-6 activation (soluble gp130) together with the vaccine and see whether it decreases the pain.
【Response】References suggested by the reviewer is humanized monoclonal antibodies or soluble gp130. In the case of vaccine adverse events, I added the last paragraph of the Discussion (line 255-258): Cytokine profiles of IL-6 and IL-10 time-courses in muscle tissues and sera reflect the regional and systemic adverse events. Grade 3 serious adverse events were observed in less than 10% of the subjects. When daily activities are disturbed, taking non-steroidal anti-inflammatory drugs will relieve the pain, suppressing the inflammatory cytokine IL-6 [30].
There are some corrections:
- In the method section you have forgotten to mention that the serum samples were also analyzed for cytokines by ELISA 【Response】Stocked serum samples were also subjected to cytokines assay.(line 100)
- Line 224-is the word "until" appropriate? 【Response】 until ⇒ within (line 246)
- In the introduction please mention how long the elders feel the pain after the vaccination of shingrix. 【Response】Following sentences are added at Introduction (line 51-54). The most common symptom was pain at the injection site and systemic adverse reactions were fatigue, myalgia, and headache. Less than 10% of the subjects experienced grade 3 reactions, but they resolved within 3 days [10].
Reviewer 3 Report
The authors made big work improving their manuscript prior to resubmission in response to comments received during 1st round of the peer review process. I have only one question left, related to the decreasing of some cytokines productions observed in muscle tissues shortly after immunization (IFN-γ on the Figure 3., IL-2 and IL-10 on the Figure 4.) It’s clearly seen there that level of IFN-γ, IL-2 and IL-10 production detected in “pre” samples, considered by authors as baseline, is much higher than detected in samples “Day1” (and “Day3” for IFN-γ and IL-2) from the control animals received the injection of the saline solution. Does it mean that injection of the saline has even stronger suppressing effect on the cytokines production than studied vaccine, or there is another explanation?
Author Response
Thank you for the valuable comment.
【Comment】The authors made big work improving their manuscript prior to resubmission in response to comments received during 1st round of the peer review process. I have only one question left, related to the decreasing of some cytokine’s productions observed in muscle tissues shortly after immunization (IFN-γ on the Figure 3., IL-2 and IL-10 on the Figure 4.) It’s clearly seen there that level of IFN-γ, IL-2 and IL-10 production detected in “pre” samples, considered by authors as baseline, is much higher than detected in samples “Day1” (and “Day3” for IFN-γ and IL-2) from the control animals received the injection of the saline solution. Does it mean that injection of the saline has even stronger suppressing effect on the cytokines production than studied vaccine, or there is another explanation?
【Response】I added the following sentences, referring a new paper No.30. (Line 251-254): Schülke [29] reviewed the pleiotropic regulatory effects of IL-10 on multiple cell types, resulting in limiting or terminating excess T-cell responses. High levels of G-CSF in sera on Day 1 of saline injection may influence the cytokine networks.
Round 2
Reviewer 1 Report
This reviewer has no further comments on this revised manuscript.
This manuscript is a resubmission of an earlier submission. The following is a list of the peer review reports and author responses from that submission.
Round 1
Reviewer 1 Report
The study by Nakayama et al. investigated the immune responses post-vaccination of recombinant zoster vaccine in mice. Although some background has been introduced, the overall motivation of this study is not clear. What is the logic of this study? The author should enhance the background and clearly state the reason.
The materials and method section is inadequate and thus need further modification in a detailed manner.
The results parts are misleading. What are the conclusions found in this study, which is significant compared to other studies? More importantly, tissues and serum samples other than muscles should also be investigated as controls to see the tissue-specific immune responses. The amount of the Ab production in serum is also a MUST, along with cytokines. The discussion part is not straightforward and seems not functionally enough based on the current findings.
Reviewer 2 Report
The work doesn't look like important to the field. The vaccine is so effective more that 95% , the adverse effect of pain can be a milder concern. If you can find a method to decrease the pain due to this vaccination , then it would be a good paper. Your findings are good but not enough to publish as a paper.
Reviewer 3 Report
The aim of the study described in the manuscript - elucidation of the mechanisms causing the side effects of the vaccination – is very important, because such of knowledges open the way to improve the existing vaccines and design a safe new ones. But the manuscript, and possible design of the study also needs the improvements.
The Abstract
In the last sentence of the abstract the authors mention that the discovered cytokine productions are related to the adverse events and can explain the marked effectiveness, but neither the adverse events nor the effectiveness were not described in the body of manuscript for the laboratory animals used in the study. If those statements are related to the human patients used the Shingrix® vaccine, it shall be declared more clearly.
- Materials and Methods
2.1. Study plan
The description of the study design must be improved. How many mice were totally used in the experiments? How many mice were in each experimental group? Was there any control group, receiving injection of some neutral agent like the PBS buffer or the normal saline solution? How many animals correspond to each datapoint showed on the Figures 1-4? The description of the used animals must be included: gender, weight, age, housing conditions.
2.2. Cytokine measurement.
Among the cytokines detected in the samples there are mentioned GM-CSF and TNF-α , witch are not shown in the Results chapter. What happened to those results? It is important because TNF-α is one of the key pro-inflammatory cytokines. The calibration curves of the Bio-Plex assay shall be attached to tha article as the supplementary material, because some cytokines like IL-4 on figure 2 are detected at extremely low concentrations.
2.3. Statistical analysis
What kind of software was used for statistical analysis of the obtained data? Was there used any algorithm to calculate the statistical significance of that data?
- Results.
There is number of missed data points on the figures illustrating the results: Figure 1 G-CSF - pre, Day7, 4W, Re Day5, Re 2W; Figure 2 IL-5 – Day3, Day5.
The letter abbreviations must be assigned to each chart on the multiple-charts figures.
It must be mentioned in text, in which cases the difference between the signal of experimental data points and background level was statistically significant. The p value for critical important data points shall be shown on the charts.
Why results for only three cytokines detected in serum (G-CSF, IL-5, IL-10) is showed, although in chapter Methods was described the Bio-Plex panel containing 8 cytokines? Were the others not tested, or the results were negative? It shall be explained.
Was there any adverse event observed among the experimental animals received the injections – like pain, fewer, change in behavior?
- Discussion.
Not showing of the data obtained from control group makes difficult the understanding of the nature of observed phenomena. So, the very short-term elevation of the pro-inflammatory cytokines production can be explained by the traumatic effect of the high-volume injection. In number of the guidelines is recommended do not exceed the 50 µl volume limit for the intramuscular injections in mice : 1) http://www1.udel.edu/research/pdf/Intramuscular-Injection#:~:text=*Mouse%3A%20IM%20injections%20are%20not,%2C%20lameness%2C%20and%20possible%20paralysis ; 2) https://iqconsortium.org/images/LG-3Rs/IQ-CRO_Recommended_Dose_Volumes_for_Common_Laboratory_Animals_June_2016_%282%29.pdf ; 3) https://s3.amazonaws.com/rfums-bigtree/files/resources/dosing-volumes-needle-minipump-guidelines-2018.pdf . Alicia M Gehling in the article “Evaluation of Volume of Intramuscular Injection into the Caudal Thigh Muscles of Female and Male BALB/c Mice (Mus musculus)” J Am Assoc Lab Anim Sci. 2018 Jan; 57(1): 35–43. https://www.ncbi.nlm.nih.gov/pmc/articles/PMC5875096/ describes the damage of the tissues caused by intramuscular injections of 100 µl solution, the same as authors of the manuscript were done. Only the comparing between the experimental and the control group can prove the specificity of the observed effects.
If the decreasing of the cytokines IL-12, IFN- γ and IL-2 production in muscle after vaccine injections (showed on the Figure 3) is connected to the increased production of IL-10, as authors claim, this hypothesis must be verified by correlation analysis. The reference to the similar effect of alum adjuvant looks unconvincing, because the tested Shingrix® vaccine do not contain the alum. On the contrary, there is the article by He L, et al. Int Immunopharmacol. 2021. PMID: 34634689 “Immune response of C57BL/6J mice to herpes zoster subunit vaccines formulated with nanoemulsion-based and liposome-based adjuvants”, where is shown that using liposome/QS-21/MPL adjuvant (AS01-like adjuvant) leads to a significant increase in the secretion of interferon gamma (IFN-γ), IL-2, IL-4, and IL-10 and showed a Th1-biased immune response in mice. Possible, there is the difference between systematic and local effects of the used immunogen. But again, the experimental results must be compared with the control group data.
In the last paragraph of the Discussion the authors wrote that “No meaningful change in Th1 cytokines of IL-2, IFN-γ, or IL-12 was observed”. If it’s related to the cytokines detected in the muscles that claim is in contradiction with obtained results, but if it’s related to cytokines detected in the serum why the data not shown in Results chapter?